# Self-Consistent Schrödinger-Poisson Study of Electronic Properties of GaAs Quantum Well Wires with Various Cross-Sectional Shapes

**DOI:** 10.3390/nano11051219

**Published:** 2021-05-05

**Authors:** John A. Gil-Corrales, Juan A. Vinasco, Adrian Radu, Ricardo L. Restrepo, Alvaro L. Morales, Miguel E. Mora-Ramos, Carlos A. Duque

**Affiliations:** 1Grupo de Materia Condensada-UdeA, Instituto de Física, Facultad de Ciencias Exactas y Naturales, Universidad de Antioquia UdeA, Calle 70 No. 52-21, Medellín 50011, Colombia; jalexander.gil@udea.edu.co (J.A.G.-C.); juan.vinascos@udea.edu.co (J.A.V.); alvaro.morales@udea.edu.co (A.L.M.); 2Department of Physics, “Politehnica” University of Bucharest, 313 Splaiul Independenţei, 060042 Bucharest, Romania; radu@physics.pub.ro; 3Universidad EIA, Envigado 055428, Colombia; ricardo.restrepo@eia.edu.co; 4Centro de Investigación en Ciencias-IICBA, Universidad Autónoma del Estado de Morelos, Av. Universidad 1001, Cuernavaca 62209, Morelos, Mexico; memora@uaem.mx

**Keywords:** quantum-well wires, self-consistent study, shape effects, electronic structure, finite elements method

## Abstract

Quantum wires continue to be a subject of novel applications in the fields of electronics and optoelectronics. In this work, we revisit the problem of determining the electron states in semiconductor quantum wires in a self-consistent way. For that purpose, we numerically solve the 2D system of coupled Schrödinger and Poisson equations within the envelope function and effective mass approximations. The calculation method uses the finite-element approach. Circle, square, triangle and pentagon geometries are considered for the wire cross-sectional shape. The features of self-consistent band profiles and confined electron state spectra are discussed, in the latter case, as functions of the transverse wire size and temperature. Particular attention is paid to elucidate the origin of Friedel-like oscillations in the density of carriers at low temperatures.

## 1. Introduction

Quantum wires (QWs) are low-dimensional semiconductor structures with strong two-dimensional localization of charge carriers, leaving a single spatial direction for their “free” displacement so that the term quasi-one-dimensional systems became adopted. This feature leads to the quantization of the energy spectrum for the motion along the cross-section of the conduction channel. The beginnings of research on this type of nanosystems date back to the 1980s, with illustrative examples in References [1,2,3,4,5,6,7,8,9,10,11,12,13,14,15,16,17], and has continued until the present, with a significant number of experimental and theoretical reports in the literature. Among the most recent works on QWs, we can mention those appearing in References [18,19,20,21,22,23,24,25,26,27,28].

With time, the concept of QW has included a modification through the term “nanowire”, due to the possibility of practical realization of wire-shaped structures with finite length [29]. Among the applications of these low-dimensional semiconductor systems, one finds the case of solar cells, in which a suitable selection of nanowire geometry can have advantages in terms of cell performance and efficiency [30]. On the other hand, in a recent report, Jia and collaborators review the state-of-the-art concerning applications of nanowires to electronics [31]. The authors discuss nanoscale devices and integrated circuits assembled from nanowire building elements, including nanowire thin-film transistors, oriented to the fabrication of high-performance large-area flexible electronics.

The self-consistent (SC) method has been a choice for determining the spectrum of electron and hole states in low-dimensional semiconductor nanostructures when many-body contributions on the energy band profile are taken into account. A typical procedure of this kind is the one described in [32], which uses the finite element method for the SC non-linear problem of coupled Schrödinger and Poisson equations for layered heterostructures with arbitrary doping profiles and layer geometries within a multiband $\vec{k}\cdot \vec{p}$ framework. In the particular case of QW-like systems, it is possible to mention initial works by Laux and co-workers [33,34]. In the first of these two works, the electron states in narrow gate-induced channels in a one-dimensional Si conduction channel are self-consistently determined solutions. In the second one, the solution of the Schrödinger-Poisson (SP) system of equations allowed to calculate electron states in a split-gate quasi-one-dimensional GaAs/AlGaAs heterostructure. Later on, Luscombe et al. solved the SCSP problem to investigate the electron confinement in laterally confined cylindrical QWs [35]. The Fourier expansion method was the approach selected by Tadić and collaborators to investigate the SC electronic structure of rectangular free-standing quantum wires [36]. A report by Trellakis and Ravaioli discusses the computational issues in the SC simulation of the electronic features of QWs [37]. The authors discuss the numerical solution for the coupled SP equation system (including exchange and correlation effects) and outline an iteration procedure—based on the predictor-corrector method—for convergence of the outer iteration.

The SC method of 2D finite differences for solving the SP equations for etched mesa GaAs/AlGaAs structures has been reported by Snider et al. in a 1990 paper [38]. This was one of the first studies showing that, for quantum wire behavior to occur, it is necessary to precisely control the width variation of the fabricated wire. The coupled Schrödinger and Poisson equations, within the Hartree approximation, have been solved by Proetto for a GaAs quantum wire with cylindrical symmetry [39]. The charge distribution, potential profile and electronic structure dependence on the wire radius and surface states’ concentration were discussed. Kerkhoven et al. have demonstrated the importance of self-consistency for analyzing the electrons confined in quantum wires [40]. By solving the Schrödinger and Poisson equations, they simulated the behavior of the low-temperature electrons behavior in QW-like structures formed by crossing layers of GaAs and AlGaAs. By using an SC screening scheme, Hu and Das Sarma have calculated the elastic mean free path of impurity-scattered carriers in a quantum wire [41]. They discussed the scale over which the electronic conduction in quantum wires is expected to exhibit metallic behavior. By solving the Poisson and Schrödinger equations iteratively for a QW-like split-gate heterojunction, Martorell et al. have studied the accuracy of the commonly used 2D approximation when applied to whole 3D systems [42]. Their work focused on interpreting general trends rather than on some specific system and simplifying assumptions for reducing the computational effort. Aristone and Sanchez-Dehesa have used the SC-SP method to investigate arbitrary profile quantum wires [43]. They discussed the numerical results for QWs of rectangular cross-sections, emphasizing the conditions under which such low-dimensional systems exhibit bi-stability. May et al. have performed an SC two-dimensional calculation of the electronic width of quantum wires formed by local oxidation on GaAlAs heterostructures [44]. They envisioned the key role that was going to be played by the SC simulations in designing novel structures and better understanding the electrostatic action of the electronic gates. Chuen et al. have calculated by an SC approach the induced charge density, capacitance, and conductance of a quantum wire [45]. They discussed the dependence of these quantities on the Fermi energy and the frequency of the external voltages.

Sharma and Suryanarayana presented a cyclic and helical symmetry-adapted formulation and large-scale parallel implementation of real-space Kohn-Sham density functional theory for one-dimensional nanostructures, with application to the mechanical and electronic response of carbon nanotubes subject to torsional deformations. They developed a high-order finite-difference parallel implementation capable of performing accurate cyclic and helical symmetry-adapted Kohn-Sham calculations in both the static and dynamic settings. Their findings were in good agreement with experimental observations and measurements. Their numerical tools and formalism were previously applied to the study of band structure and bending properties of large *X* (X= C, Si, Ge, Sn) nanotubes and *X*ene sheets [46,47].

A theoretical study of two-dimensional quantum dots with the shape of a regular polygon where the number of sides has varied from three (triangle) to infinity (circle) has been reported by Popescu and coauthors [48]. They used the finite element method to calculate the energies and probability densities for an electron confined in the quantum system. Among their findings, they report that no matter the shape, any regular polygonal quantum dot with more than four sides and a given area has just the same quantitative optoelectronic properties. Additionally, they found that at the nanoscale, the shape may not be as important as the size is.

Efforts to develop cheap and efficient schemes for the electron states’ numerical solution in wire-shaped nanosystems have continued throughout the years. A 3D finite-difference time-domain simulation was recently used to solve the problem in cylindrical QWs [49]. Bearing all this in mind, we are interested in bringing a new vision to the subject by solving the SCSP problem in quantum wires with the help of finite-element approach. We shall explore the influence of the QW cross-section shape by considering the circle, square, triangle, and pentagon geometries. We aim at discussing the features of confined state energies and electron density in the system. We are also widely motivated to extend the work of Popescu et al. [48] to analyze to what extent the shape of the QW boundary (which is controlled by the number of sides of the regular polygon corresponding to its cross-section) can or not be relevant compared to other characteristics of the system such as the cross-sectional area, the density of donors with which the system is doped, and the temperature. In this article, using the effective mass approximation, we study an electron confined in a GaAs QW of infinite length and infinite confinement potential added to an electrostatic potential at the boundary. These two potentials are associated with QWs surrounded by a vacuum or by a matrix with an energy gap greater than that corresponding to the QW one. We report the energy levels in each of the structures with different cross-sections as a function of the transversal area, the doping donor density, and the temperature. We analyze the oscillations that appear in electron density at low temperatures and discuss the contribution made by each of the confined electron states to such oscillations. The article is organized as follows—Section 2 presents the theoretical framework of the simulation. In Section 3 we show and discuss the results obtained, whilst Section 4 is devoted to the conclusions of the work.

## 2. Theoretical Framework

The system of interest corresponds to a GaAs QW with a so-called exposed lateral surface. Under this condition, the model has a core-shell structure, where the core constitutes the wire region and the shell would be the surface of the semiconductor. The electronic properties are studied for this system with four different cross-section geometries. In Figure 1 are depicted the four structures considered in this work and are defined the coordinate axis, the surrounding material, and the radius of the cylindrical wire case. The wire length is large enough such that a wire with infinite length along the *z*-axis is a good approximation. Due to the vacuum surrounding matrix, the confinement potential is zero inside the wire region and infinite elsewhere. Having the system with exposed borders is a characteristic of fundamental importance in the numerical approach since the boundary conditions are established for the potential V(Ω), where Ω represents, in each case, the boundary. This function *V* establishes the form of the confinement potential.

In the GaAs quantum wire with the exposed surface, a depletion region is generated at the bottom of the conduction band, which allows the appearance of empty states above the Fermi level; these are surface states that cause a charge transfer from bulk energy states to surface energy states. Consequently, a phenomenon known as “Fermi level Pinning” is present [50]. That is, the Fermi level is kept fixed to a characteristic value within the bandgap of the system, and this value is independent of the density of donors or acceptors. For GaAs, it is normally used the value on the surface which corresponds to half of the energy bandgap. So, defining the Fermi level as $E_{F}$, then the potential at the frontier for each structure will be $V^{*}\left(\Omega \right)-E_{F}=0.7$ eV [35].

The Schrödinger-Poisson Equation multiphysics interface, available in the COMSOL-Multiphysics version 5.4 [51,52,53], creates a bidirectional coupling between the electrostatics interface and the Schrödinger Equation interface to model charge carriers in quantum-confined systems. The electric potential from the electrostatics contributes to the potential energy term in the Schrödinger equation. A statistically weighted sum of the probability densities from the energy eigenvalues of the Schrödinger equation contributes to the space charge density in the electrostatics. All spatial dimensions (1D, 1D axial symmetry, 2D, 2D axial symmetry, and 3D) are supported.

In the numerical procedure, it is required to solve coupled SP differential equations in a self-consistent way to obtain the potential profile and electron density for each of the quantum wire systems and the corresponding wave functions and energy eigenvalues [32]. Here, we use the effective mass approximation and choose the finite element method (FEM) to perform the SCSP calculation. It is worth mentioning here that this is the typical SC scheme of working within a single-band or multiband $\vec{k}\cdot \vec{p}$ environment in semiconductor physics. However, there are very recent reports on the adaptation of density functional theory which is, intrinsically, self-consistent to deal with nanowire-type systems or with nanotubes; thus opening a way to a powerful, although more computationally demanding, microscopic calculation tool for this particular kind of systems [46,47].

Following [54], we select the electron gas density given by the Thomas-Fermi approximation as the starting point for the method. This quantity will enter the Poisson equation to contribute to the system’s charge density along with donor concentration $N_{d}$ (assumed to be fully ionized) within the QW core. We have
(1)$$n\left(x,y,T\right)=N_{C}\;F_{1/2}\left(\frac{E_{F}-V^{*}\left(x,y,T\right)}{k_{B}T}\right)\;,$$
where $N_{C}={\left(2m^{*}k_{B}T/\pi \hbar^{2}\right)}^{3/2}/4$ is the effective density of states, $V^{*}\left(x,y,T\right)$ is the electronic potential generated by the Fermi level Pinning on the exposed lateral surface, the level of doping and the lateral dimensions of the system, $m^{*}$ is the electron effective mass, and $F_{1/2}\left(x\right)$ is the Fermi-Dirac integral given by
(2)$$F_{1/2}\left(\xi \right)=\frac{1}{\Gamma (3/2)}\int_{0}^{\infty }\frac{\beta^{1/2}}{e^{\beta -\xi }+1}\;d\beta \;.$$

In this equation, Γ is the Gamma function and in this case $\xi =\left(E_{F}-V^{*}\left(x,y,T\right)\right)/k_{B}T$, where $k_{B}$ is the Boltzmann constant and *T* is the temperature. The potential energy of the electron is given by $V^{*}\left(x,y,T\right)=-e\;\phi \left(x,y,T\right)$, where ϕ(x,y,T) is the Hartree potential. Then, the associated Poisson equation is:(3)$$\nabla^{2}\phi \left(x,y,T\right)=-\frac{1}{\varepsilon_{0}\;\varepsilon_{r}}\;\rho \left(x,y,T\right)\;,$$
where $\rho \left(x,y,T\right)=e\;[N_{d}-n\left(x,y,T\right)]$ is the charge density, *e* is the electron charge, and $\varepsilon_{r}$ and $\varepsilon_{0}$ are the GaAs relative permittivity and vacuum permittivity, respectively. This equation must be solved taking into account the boundary conditions imposed by the Fermi level pinning, which for GaAs takes the form $\phi \left(\Omega \right)=-(E_{F}+0.7$ eV)/e.

The potential, ϕ(x,y,T), obtained through Equation (Equation 3) must contribute to the potential energy term in the Schrödinger equation as −eϕ(x,y,T). On the other hand, the electrons are assumed to be totally confined within the volume of the QW and, therefore, it must be satisfied that for the Ψ-electron wave function the condition Ψ(Ω)=0 must be satisfied. Under these considerations, the Schrödinger equation for the confined electron in the QW reads
(4)$$-\frac{\hbar^{2}}{2m^{*}}\nabla^{2}\Psi \left(\vec{r}\right)+\left(V-e\;\phi \left(x,y,T\right)\right)\Psi \left(\vec{r}\right)=E\;\Psi \left(\vec{r}\right),$$
where *V* is the confinement potential of the structure. Considering the infinite length along the *z*-axis, the 3D wave function of the system can be written as
(5)$$\Psi \left(\vec{r}\right)=e^{ik_{z}z}\;\psi \left(x,y\right)\;.$$

By using Equation (Equation 5) in Equation (Equation 4), we obtain the following 2D differential equation
(6)$$-\frac{\hbar^{2}}{2m^{*}}\nabla^{2}\psi \left(x,y\right)+\left(V-e\;\phi \left(x,y,T\right)\right)\psi \left(x,y\right)=E_{0}\;\psi \left(x,y\right)\;,$$
where $E=\frac{\hbar^{2}\;k_{z}^{2}}{2\;m^{*}}+E_{0}$.

As the electrons must be confined to the interior of the quantum wire, therefore, to solve this equation, it is necessary to impose the boundary condition ψ(Ω)=0. Solving the last equation, we shall find the first set of self-functions $\psi_{i}$ and self-energies $E_{0,i}$ for the system. With these elements at hand, it is possible to calculate the electron density associated with the occupation of each of these states:(7)$$\eta \left(x,y,T\right)=\sum_{i=1}N_{i}\;F_{-1/2}\left(\frac{E_{F}-E_{0,i}}{k_{B}T}\right){\left|\psi_{i}\left(x,y\right)\right|}^{2}\;,$$
where $N_{i}=g_{i}\sqrt[{3}]{4N_{C}}$, $E_{F}$ is the Fermi energy and $g_{i}$ is the degeneracy factor. This equation represents the density of electron gas in a (x,y)-point at a temperature *T*. It can also be understood as the summation of the probability of finding the electron in the point (x,y) inside the quantum wire in each determined state $\psi_{i}$ with energy $E_{0,i}$. From the electron density calculated in Equation (Equation 7), a new profile for the charge density of the system is obtained:(8)$$\rho_{new}\left(x,y,T\right)=e\left[N_{d}-\eta \left(x,y,T\right)\right]\;.$$

By solving the Poisson Equation (Equation 3) and using the corresponding charge density profile, a new Hartree potential $\phi_{new}\left(x,y,T\right)$ is obtained that will -again- contribute to the potential energy term in the Schrödinger equation. Then, via the solution of Equation (Equation 6), with this consideration, we obtain a new set of eigenfunctions and eigenvalues for the system $\psi_{i}^{new},E_{0,i}^{new}$. This set will be associated with a new electron density profile $\eta_{new}$ relative to each state of the system’s occupation. In this way, the process is repeated iteratively until the absolute value of the difference between potential terms corresponding to two successive self-consistent steps is smaller than a certain tolerance $|U-U_{old}|<10^{-6}$ eV, where U=V−eϕ(x,y,T). At this point, the system has reached self-consistency, finally obtaining a set of eigenstates, eigenvalues, a definitive form for the potential profile of the system, and the SC electronic densities.

## 3. Results and Discussion

For all calculations, the following parameters have been set—effective mass of electron in GaAs $m^{*}=0.067\;m_{0}$ (where $m_{0}$ is the mass of the free electron) and dielectric constant $\varepsilon_{r}=12.9$. All the equations have been solved through the finite element method with the COMSOL-Multiphysics licensed software (5.4, COMSOL AB, Stockholm, Sweden) [51,52,53]. The used typical numerical parameters are inner mesh with triangular-shaped elements, 6550 elements, 160 edge elements, 3356 mesh vertices, 40 as the maximum number of iterations of the self-consistent method, and 10${}^{-6}$ as the absolute tolerance.

Figure 2 shows the plots of the first five QW confined state wave functions in each of the four configurations studied, from top to bottom—circle, square, triangle and pentagon. It should be noted that the cross-sectional area for all considered QWs has been kept the same. The cross-sectional area of all structures are set to be equal to $\pi \;R_{0}^{2}$ (the area of the circular QW with $R_{0}$ radius). The electron density $N_{d}$ has been fixed as $3\times 10^{19}$ cm${}^{-3}$ for all cases. For all considered structures, the $\psi_{1}$ and $\psi_{2}$ states are doubly degenerated. The states $\psi_{3}$ and $\psi_{4}$ are degenerate only for circular and pentagon QW. The states’ sequence of degeneration in the structures shown is as follows—(1,2,1,2,2) for the square and triangle and (1,2,2,1,2) for the circle and pentagon. The color scale in each figure goes from blue, which corresponds to the wave function’s negative values, to dark red, which represents positive values of the wave function. The yellow color indicates the points at which the wave function is zero. The first column on the left corresponds to the ground state $\psi_{0}$ for each system. There, it is possible to notice the *s*-type character that this state acquires for all studied configurations, as detailed by the next paragraph. Additionally, it is emphasized that the electrons in the wire with the triangular section are more confined towards the symmetry axis than in the other structures. The first and second excited states, $\psi_{1}$ and $\psi_{2}$, are presented on the second and third columns from left to the right. Note that these states have a *p*-type character. Finally, in the rightmost two columns, the third and fourth excited states, $\psi_{3}$ and $\psi_{4}$, are shown for each configuration. These states show a *d*-type behavior, as can be noticed from their projections.

The xy-cross-sections of the electron wave functions exhibit particular symmetries that we refer to by partial analogy with electronic orbitals in atoms. For all types of wire, the ground level orbital is the *s*-like orbital, which means a single central extreme (*sharp* orbital) of the wave function in the xy-plane, corresponding to the symmetry center of the wire cross-section. There is only the central lobe (positive amplitude) and no angular nodal surface (zero amplitude) for the *s*-like orbitals. The second energy level (second and third column) is double degenerate in all types of wire. The two orthogonal wave functions corresponding to this energy have two extremes each (*principal* orbitals), therefore being denoted as *p*-like orbitals. They both have one lobe of positive amplitude and one lobe of negative amplitude, separated by an angular nodal surface perpendicular to the xy-plane. The angular nodal surfaces of the two different *p*-like orbitals are orthogonal.

Figure 3 shows the self-consistent confinement potential for each of the structures. In the calculation, the circular QW radius has been taken equal to 50nm, which fixes the cross-sectional area of all the other systems. Keeping the color code in which red indicates the most significant values shows that the potential becomes higher at the boundary regions, which favors electronic confinement in the core regions within the QW. Electrons feel a similar potential near the symmetry axis of each structure. The change is noticeable mostly near the border of the QW where the potential presents a less smooth behavior.

Figure 4 shows the electron density η(x) (with y=0) for each structure, normalized to the doping value $N_{d}=3\times 10^{19}$ cm${}^{-3}$. It has been calculated along the arrow indicated in the inset at the bottom of each figure. Accordingly, a higher concentration is evidenced in the core region or near the symmetry axis of each QW. This feature was expected, given that—as seen from Figure 3—the confining potential is low around this zone. The results plotted in Figure 4 are for T=10 K. In these systems, with exposed borders, at low temperatures, Friedel-like spatial oscillations of the carrier density appear [55]. These can be viewed as irregularities in the density profile, especially around the central zone of the QW. These oscillations could be explained by the presence of subbands associated to surface states and by how the electrons populate each of them. They can be caused by the electronic occupation of the oscillation-related lower states closer to the center of the QW, while the higher occupied states can also contribute to a lesser extent. One may observe that very fast decrease of the electron density occurs in regions close to each system’s boundary, while in regions close to x=0 a maximum value is reached. However, $\eta \left(x\right)/N_{d}$ does not reach 1.0 in any system. This is evidence of the charge transfer from inner states (the core) to surface states since the complete system must remain neutral. These states are the ones that contribute to the Fermi level pinning.

Figure 5 presents the ground state energy variation for the four studied GaAs QWs as a function of $R_{0}$. In this figure, the donor density has been taken as $N_{d}=2\times 10^{18}$ cm${}^{-3}$, the temperature is T=10 K, and the Fermi level is $E_{F}=0$. As long as $R_{0}$ increases for all structures, a clear decreasing behavior of the ground state is evidenced [56,57]. This decrement is more abrupt for the circular system, and it is less so for the triangular one. To explain this fall in energy, we must notice that, due to the increase in $R_{0}$, the cross-section area of all the QWs augments leads to a reduction of the electronic confinement, a redshift in the eigenvalues takes place. For $R_{0}=10$ nm, the ground state energy is very similar in all structures, taking a value around 0.67 eV. Something alike occurs for $R_{0}=50$ nm where the energy decreases to approximately −0.075 eV. For intermediate values, a separation between the states corresponding to each structure is clearly noted, this separation being greater for $R_{0}=30$ nm. This fact appears in the inset, where the difference between the ground state of the triangular and circular GaAs QWs is shown. It reaches a maximum value of approximately 130 meV. The ground state’s behavior for the circular, square, and pentagonal systems is very similar, and the largest differences appear for the triangular system.

Figure 6 shows the first excited state energy for the four GaAs QWs as a function of the $R_{0}$ under the same conditions as for Figure 5. We can see that a clear decreasing trend is evidenced as $R_{0}$ increases. The first excited state of the four systems presents an energy of 0.69 eV at $R_{0}=10$ nm which decreases to −0.06 eV when $R_{0}=50$ nm. Quantitatively, this first excited is very similar for the circular, square, and pentagonal QWs. The main difference appears with the triangular system, which has higher energy. This behavior is evidenced in the inset where the circular and triangular systems’ energy difference is presented. As in Figure 5, a maximum difference is noted for $R_{0}=30$ nm.

In Figure 7, the first two energy states are shown for each of the systems—circular (a), square (b), triangular (c), and pentagonal (d)—plotted as functions of the $R_{0}$ parameter. For $R_{0}<25$ nm the systems show an approximately parabolic decrease in energy. Then, for 25 nm$<R_{0}<35$ nm this decrease displays an approximately linear behavior and for $R_{0}>35$ nm the decrease becomes approximately exponential. There is a degeneracy of order two for all systems concerning the first excited state (this generation is indicated by label 2 in each figure).

Figure 8 shows the energy difference between the first excited state and the ground state for each of the four GaAs QW structures as a function of the $R_{0}$ parameter. In this case, the donor density and temperature have been kept fixed at $N_{d}=2\times 10^{18}$ cm${}^{-3}$ and T=10 K, respectively. An increase in the separation between these two states is evidenced as $R_{0}$ augments within the range between 10 nm and 17 nm, going from an average value of 28 meV to around 38 meV, for all structures. In this range, it is also noted that the separation between levels in the cases of square and pentagonal geometry presents a very similar behavior. The curves show a linear and parallel tendency. The greatest separation of $E_{1}-E_{0}$ appears from the comparison between the circular and triangular QWs, followed by that involving the pentagonal and triangular wires. Within the range 17 nm $<R_{0}<30$ nm, the separation between the levels does not show significant growth, maintaining an average value of about 38 meV. In this region, the curves show a kind of oscillatory behavior, presenting a close approach between the circular, square and triangular QWs for a value of $R_{0}∼23.5$ nm. Then, for values $R_{0}>30$ nm, these energy values keep getting closer until reaching a difference of 8.8 meV approximately at $R_{0}=50$ nm for circular and pentagonal QWs and of 10 meV and 12.3 meV for wires with square and triangular sections, respectively. Note the similar behavior shown by the curves corresponding to the circular and pentagonal QW for values of $R_{0}$ greater than 35 nm. This is explained as a consequence of the fact that electrons, for a large QW cross-section, do not feel the edge effects in the system. This fact can be analyzed in conjunction with the results presented in Figure 4, which show that the electron density is practically concentrated at the center of the structure, and the pentagonal profile is closer to the circular one. The opposite occurs with the square and triangular systems, for which the profile electron density is changed in greater extent due to the shape of system boundaries. In the region, $R_{0}<17$ nm, a greater separation between the curves occurs again for the circular and triangular QWs. But now, the curves have been inverted, being higher than that of the triangular QW.

In previous figures, the electronic energies in each system have been plotted as functions of $R_{0}$, keeping the doping fixed at $N_{d}=2\times 10^{18}$ cm${}^{-3}$ and the temperature at T=10 K. Next, the variation of the energies will be studied while allowing variations of the $N_{d}$ parameter and keeping fixed the dimensions of the QWs and the temperature. At this point, it must be remembered that the system under study is a QW with exposed boundaries. Thus, an immediate consequence is the Fermi level pinning due to the transfer of charge towards surface states. These states’ appearance is transcendental since it fixes the surface potential and the Fermi level independently of the donor density in the system.

Figure 9 shows the ground state energy for each of the GaAs QW systems as a function of the donor density, $N_{d}$. The remaining system parameters have been fixed at $R_{0}=30$ nm and T=10 K. Note that for $N_{d}<5\times 10^{18}$ cm${}^{-3}$, there is practically no difference in electron energy when comparing results for the four distinct cross-section geometries. This fact can be more clearly noticed by seeing the upper inset, in which the energy difference between the ground state of the triangular and circular QWs is presented. One may observe that the difference between state energies is of the order of 4 meV when $N_{d}\to 0$. On the other hand, when $N_{d}=5\times 10^{18}$ cm${}^{-3}$, this difference reaches a maximum value of 18.7 meV to subsequently show an approximately constant decrease around 7.5 meV. Note the type of linear decrease that occurs for $N_{d}>12\times 10^{18}$ cm${}^{-3}$ for all structures. In general, the ground state shows a decrease from 0.69 meV for $N_{d}=0$ to approximately −0.5 meV, independently of the QW shape. The lower inset is a magnification of the curves, where the very close behavior for circular, square and pentagonal QW structures is again obvious, with the state of the triangular system presenting a more noticeable separation.

Figure 10 presents the first two electronic states for each of the GaAs QW systems as a function of the donor density, $N_{d}$. The temperature has been kept fixed at T=10 K and the geometric $R_{0}$ parameter at 30 nm. For all the structures, a decreasing behavior is observed with the increase in $N_{d}$, this decrease being approximately linear for values $N_{d}>10^{19}$ cm${}^{-3}$. It should be noted that there is a degeneracy of the second degree in the case of the first excited state for all systems. Just to quantify the fall in energy levels, for the circular system it goes from 0.7 eV for $N_{d}=10^{16}$ cm${}^{-3}$ to −0.5 eV for $N_{d}=3\times 10^{19}$ cm${}^{-3}$ in the case of the first excited state.

We are presenting in Figure 11 the difference between the first excited state energy and the ground state energy for all considered GaAs QW systems, depicted as a function of the electron density, $N_{d}$. The same parameters as in Figure 10 have been kept fixed. A monotonically decreasing behavior is immediately evident for all curves. In the same way it should be noted that the maximum separation between these two levels takes place in the particular case of the QW with triangular cross-section, reaching a maximum value of approximately 32.2 meV at $N_{d}=5\times 10^{18}\;$ cm${}^{-3}$, and decreasing to 18 meV at $N_{d}=3\times 10^{19}$ cm${}^{-3}$. In order of separation between these two levels, the next structure is the square-shaped QW which goes from 27.2 meV to 15.7 meV in the same range of $N_{d}$. Here, a particular behavior is present for circular and pentagonal QWs. Actually, they present exactly the same separation between these two lowest levels for $N_{d}=5\times 10^{18}$ cm${}^{-3}$, taking a value of 24.7 meV. However, as $N_{d}$ is increased, an appreciable difference is reached between these two kinds of QWs, the circular one taking a slightly higher value. This behavior is maintained with the increase in $N_{d}$.

The variation of the lowest six energy levels for the four GaAs QW systems as a function of temperature is shown in Figure 12. For this case, $R_{0}$ has been fixed at 50 nm and $N_{d}=2\times 10^{18}$ cm${}^{-3}$. The number that appears next to some states indicates their degree of degeneracy. Note the similar behavior of the energy degeneracy for the circular and pentagonal QWs (also for the triangular and square QW). This was expected, bearing in mind the results obtained in Figure 9 and Figure 11. It should be noted that the highest value of the ground state energy happens for the triangular QW system, taking a minimum value of −72.9 meV at T=10 K and a maximum of −65.3 meV at T=290 K. Note that all states show an increasing trend with rising temperatures. This fact is a consequence of the stronger confinement at high temperatures.

Figure 13 shows the difference between the first excited state energy and the ground state energy as a function of temperature for the four QW types analyzed. The other parameters have been kept fixed in a similar way as in Figure 12. The highest separation is given for the triangular system, and goes from 12.3 meV at T=10 K to 13.1 meV at T=290 K. It is followed by the square QW which, for the same temperature range, goes from 9.9 meV to 10.5 meV. Finally, as in the previous figures, again the most similar behavior is exhibited by the curves of circular and pentagonal system. This was seen in Figure 12 with the similar behavior that the first two levels followed in both structures. However, this similarity is not present in the case of higher states. The minimum separation between the levels is given for pentagonal-shaped QW, which, in the same range of temperatures studied, goes from 8.6 meV to 9.2 meV. Note the increasing character of all the curves with temperature.

The results for the electron density profile given by Equation (Equation 7) are presented in Figure 14 for the circular QW at a temperature of T=10 K and $R_{0}=50$ nm. Figure 14a shows the total density (black curve). The dashed vertical lines indicate the local maxima that appear as oscillations in points (1), (2) and (3). The contributions coming from the different electron states in these points appear in Table 1. The first number in parentheses corresponds to the azimuthal quantum number (*m*) and the second to the radial quantum number (*l*). Note that in the case of the circular quantum wire, and due to its axial symmetry, the $\psi_{i}\left(x,y\right)$ wave function in Equation (Equation 6) can be written as $\psi_{i}\left(x,y\right)=f_{l}\left(\rho \right)\;\exp\left(im\phi \right)$, where m=0,±1,±2,... is the azimuthal quantum number, l=1,2,3,... is the radial quantum number, and $\rho =\sqrt{x^{2}+y^{2}}$. The percentage that appears next to each state in Table 1 corresponds to the contribution of each of them to the oscillation of the total density profile at points (1), (2), and (3) in Figure 14, where the probability density has been plotted for the states with a contribution greater than 15.0% for all peaks. As seen from Figure 14a and Table 1, for the oscillation centered at x=4.75 nm, denoted by (1) in the figure, the highest contribution is due to the ground state of the system, $\psi_{0,1}$, with 23.0%, followed by the $\psi_{0,2}$ and $\psi_{1,2}$ states with 19.6% and 19.2%, respectively. The last significant contribution is due to the $\psi_{1,3}$ state with 16.0%. Meanwhile, the other states present a contribution of less than 10.0% for this first oscillation in the electron density profile. Accordingly, this particular oscillation is due mainly to the contribution of the lower states of the system; note that states with m>3 do not contribute to the appearance of this oscillation. On the other hand, for the oscillation centered at x=12.25 nm, denoted as (2) in the figure, the state that contributes the most is $\psi_{1,1}$ with 23.5%, followed by $\psi_{2,1}$ with 16.3% and $\psi_{2,2}$ with 12.1%. The other states present contributions of the order of 10.0% and less. Note that the lower states $\psi_{0,1}$ and $\psi_{0,2}$, that for the first oscillation contributed 42.6% to the electron density, for this second oscillation only provide 10.9%. However, the $\psi_{1,1}$ and $\psi_{2,1}$ states went from 9.7% in the first oscillation to 39.8% in the second one. For the third oscillation at x=20.5nm, denoted by (3) in the figure, being more tenuous, a rather equitable contribution is evidenced between the states with the largest *m* quantum number. This contribution is on average 13.26%, while the proportion from the lowest states is less than 7.0%. An important conclusion here is that the oscillation generated in the density profile near the central zone of the circular QW is due to the contribution of the lower states of the system, while the second oscillation is caused by intermediate occupied states and the final one—around 20 nm—occurs due to the highest occupied states. It should be mentioned that the empty spaces in Table 1 correspond to states that do not contribute to the electron density at that specific point. For example, the $\psi_{0,3}$ state contributes to the oscillations presented in points (1) and (2), but it does not contribute to the oscillation generated in point (3). The curves plotted in Figure 14b correspond to the electron density for the cylindrical QW, separating the individual contribution of states with different *m*-quantum number. This quantum number is kept fixed, and the sum in the Equation (Equation 7) is made over the *l*-quantum number. It should be noted that, for m>6, there are no longer occupied states, and therefore they do not contribute to the electron density. Figure 14c, together with the total electron density (black curve), shows the sum over occupied states up to the state m=i (the $\Sigma m_{i}$ symbol represents the value of summation). Sums up to the states that have m=6—the highest occupied—are shown. Figure 14b,c are evidence that the contribution to the first oscillation’s appearance at x=4.75 nm is mainly caused by electrons occupying the states with m≤3. On the other hand, for the outermost oscillations that are located at x=12.25 nm and x=20.5 nm, the oscillations are caused by states with 4<m<6, that correspond to the highest occupied states in the system. All this means that electrons with lower state energies are located close to the center of the structure’s symmetry, between 0<x<10 nm; while electrons in higher energy states mainly locate at intermediate regions, 10<x<35 nm.

Finally, Figure 15 shows the electron density as a function of the *x*-position for the structures with square, triangle, and pentagon cross-section geometries. It is equivalent to Figure 14a for the circular system. Each figure shows the plot of the probability density of the states that contribute simultaneously to the two oscillations in all the systems of Figure 15. That is, $\psi_{0}$, $\psi_{2}$, $\psi_{5}$, $\psi_{10}$, and $\psi_{14}$, as indicated in Table 2, where the percentage contribution of each one to the electron density at the position determined by the dashed line (points (1) and (2)) is also shown. States above $\psi_{15}$ present a contribution much less than 1.0% and therefore were not included in Table 2. In these structures, we see again how the lower states provide a higher contribution to the electron density near the symmetry axis of the QW systems. Since these figures have been calculated at low temperature (T=10 K), then the electrons will be—to a greater extent—occupying the lowest states of each QW. The opposite case occurs at points far from the symmetry axis of the structure, where we see a minimum contribution from the lowest states of the system and a higher contribution—in percentage—from the highest occupied states. The x=0 coordinate corresponds to the symmetry axis of each structure. Note the asymmetry in the electron density profile concerning this point for the triangular and pentagonal systems.

## 4. Conclusions

Electronic properties such as wave functions, state energies, potentials and electron densities have been calculated in a self-consistent way, using the finite element method, for GaAs quantum wire systems of different cross-section geometry, taking into account variations in geometric parameters, such as cross-sectional area and non-geometric parameters, such as the donor density and the temperature. It has been shown that the increase in cross-section and/or donor density in all structures generates a lesser degree of confinement by the self-consistent potential and, therefore, a decrease in electronic energies. The opposite case occurs when the temperature is increased, for which there is an increase in the self-consistent potential profile, thus impacting on the increase of energy eigenvalues for all systems. The system that presents the highest values of the confined energy levels is the quantum wire with a triangular cross-section, and the one with the lowest energies is the circular wire. At low temperatures, all structures present irregularities in the electron density profile. These Friedel-like oscillations are due to the degree of occupation of internal and surface states (that arise from having the surface of the quantum wire exposed) by conduction electrons. This new understanding of the quantum wires can be extended without significant changes to the study of finite-length nanowires with the most diverse geometries. Hence, we believe that the results and scheme presented here can be of interest to researchers in the area.

## Figures and Tables

**Figure 1 nanomaterials-11-01219-f001:**
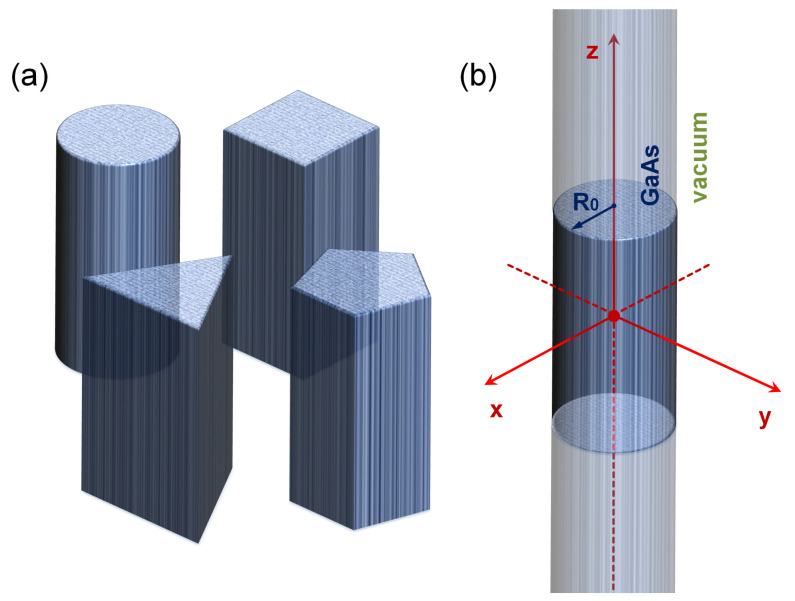
Pictorial view of the GaAs quantum well wire studied in this work. In (**a**) are depicted the four considered structures. In (**b**) are defined the coordinate axis, the $R_{0}$-radius for the cylindrical case, and the vacuum surrounding matrix. The wire length is large enough such that a wire with infinite length along the *z*-axis is a good approximation. Due to the vacuum surrounding matrix, the confinement potential is zero inside the wire region and infinite elsewhere.

**Figure 2 nanomaterials-11-01219-f002:**
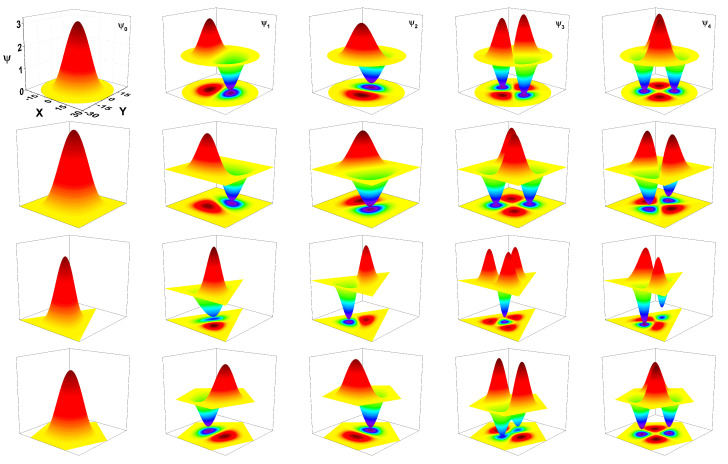
First five wave functions for a confined electron in GaAs quantum wires. The rows from top to bottom are for circle, square, triangle, and pentagon, respectively. The left-hand side column corresponds to the ground state; the next four columns from left to right are for each system’s first four excited states. For all figures, the cross-sectional areas are the same.

**Figure 3 nanomaterials-11-01219-f003:**
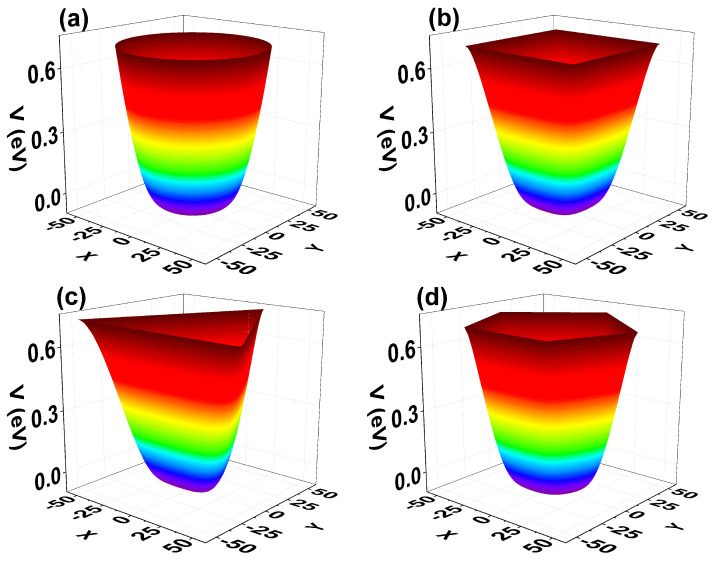
Self-consistent confining potentials for the different GaAs quantum wire geometries investigated: circle (**a**), square (**b**), triangle (**c**), and pentagon (**d**). Calculations are with $N_{d}=3\times 10^{19}$ cm${}^{-3}$ and T=10 K.

**Figure 4 nanomaterials-11-01219-f004:**
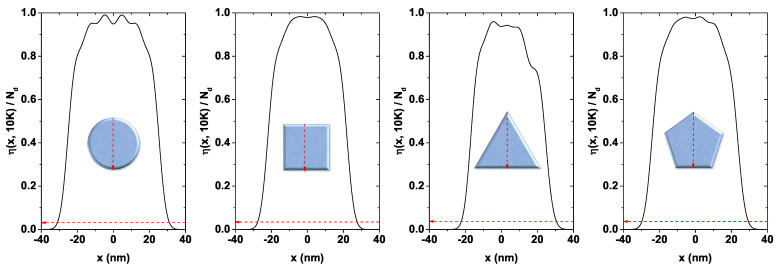
Normalized electron density functions for each investigated GaAs quantum wire structure as a function of the *x*-coordinate (see the red arrow in the inset in each figure) for T=10 K and $N_{d}=3\times 10^{19}$ cm${}^{-3}$. For all figures the cross-sectional area has been set at A=2500π nm${}^{2}$.

**Figure 5 nanomaterials-11-01219-f005:**
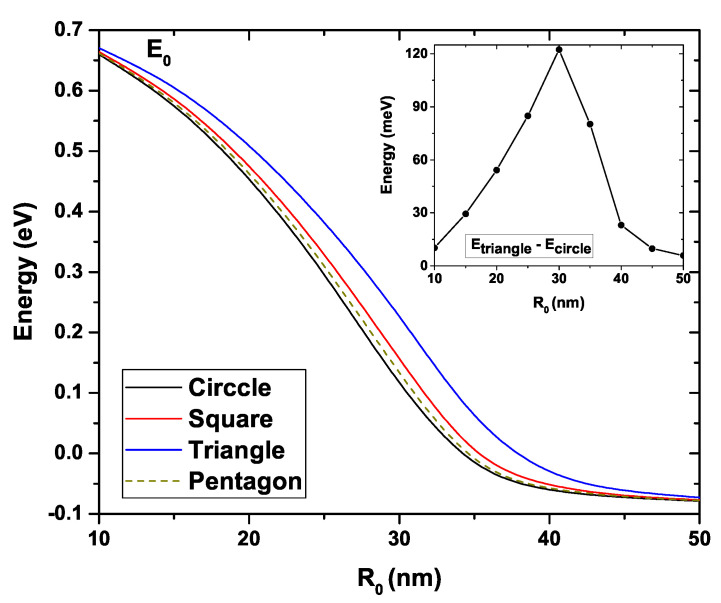
Ground state energy for each GaAs quantum wire structure as a function of the $R_{0}$ parameter. The inset shows the energy difference between the triangular and circular wires. Calculations are for $N_{d}=2\times 10^{18}$ cm${}^{-3}$, T=10 K, and $E_{F}=0$.

**Figure 6 nanomaterials-11-01219-f006:**
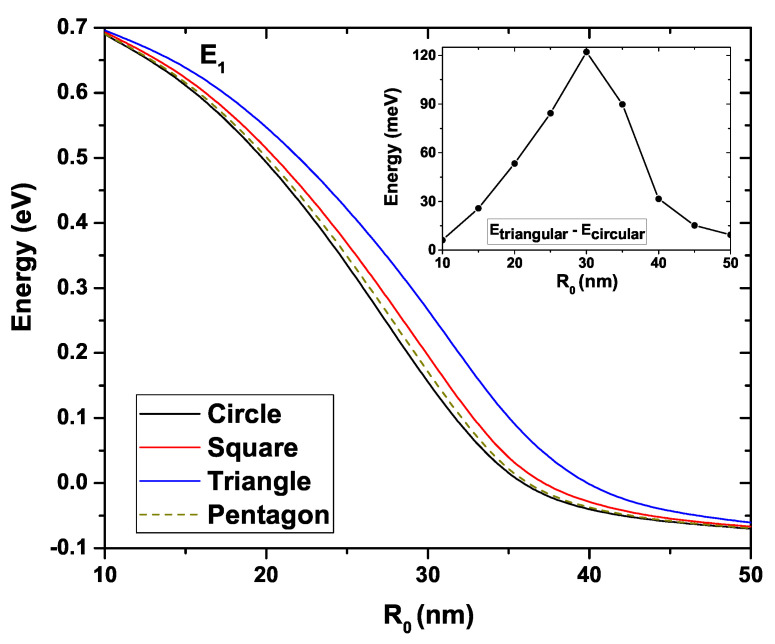
First excited state energy for each GaAs quantum wire structure as a function of $R_{0}$. The inset shows the energy difference between the triangular and circular wires. Calculations are for $N_{d}=2\times 10^{18}$ cm${}^{-3}$, T=10 K, and $E_{F}=0$.

**Figure 7 nanomaterials-11-01219-f007:**
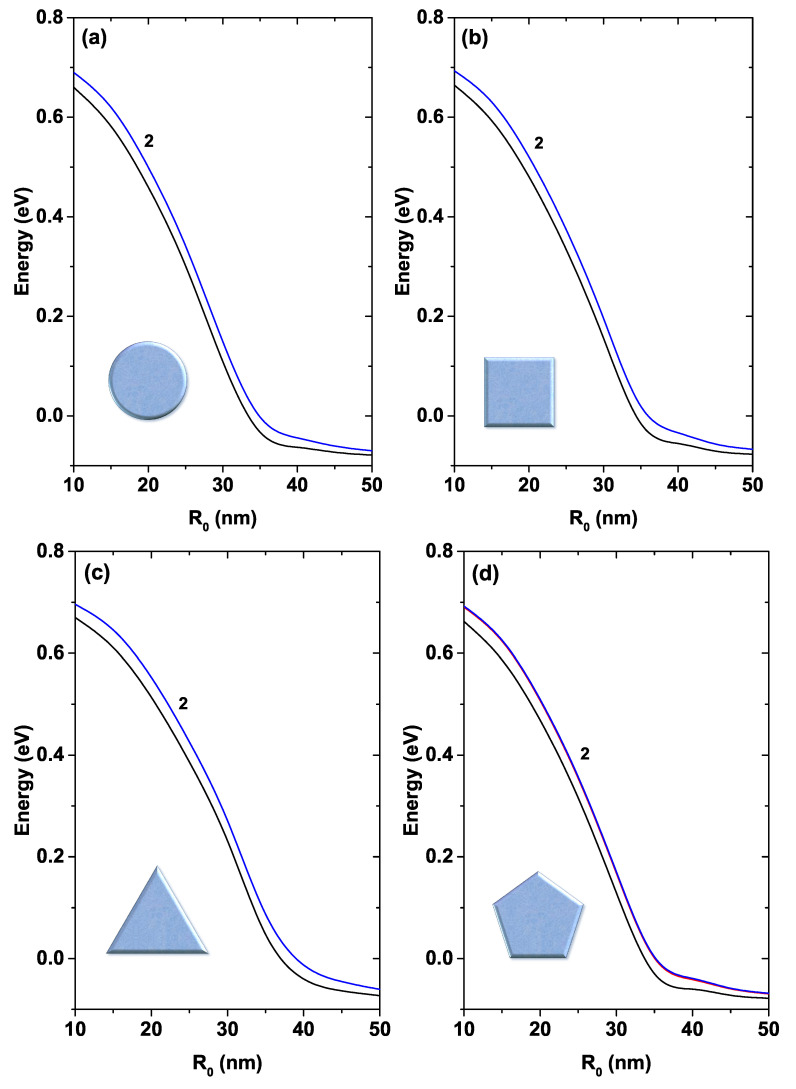
First two energy levels for a confined electron in a GaAs quantum wire as a function of the $R_{0}$ parameter. The label 2 indicates that the first excited state is doubly degenerated. Calculations are for $N_{d}=2\times 10^{18}$ cm${}^{-3}$, T=10 K, and $E_{F}=0$.

**Figure 8 nanomaterials-11-01219-f008:**
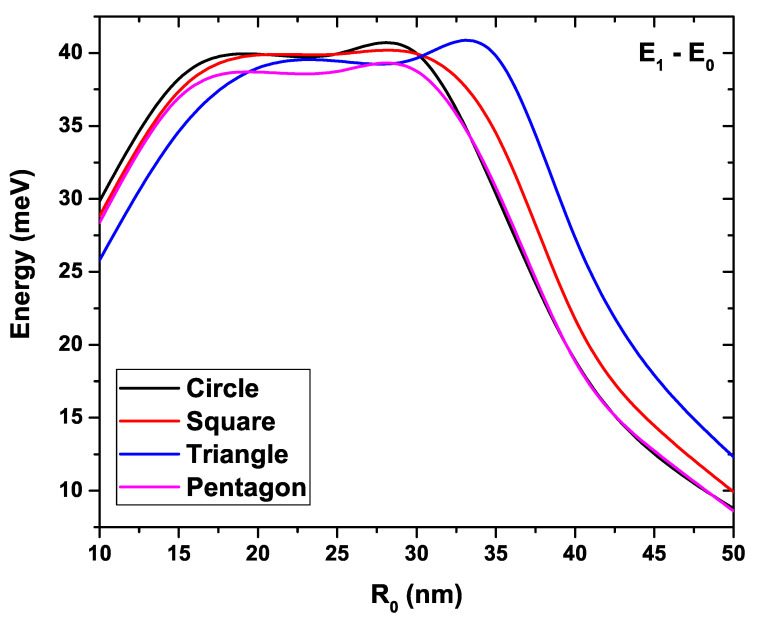
Energy difference between the first excited state and the ground state for each GaAs quantum wire as a function of the $R_{0}$ parameter. Calculations are with $N_{d}=2\times 10^{18}$ cm${}^{-3}$, T=10 K, and $E_{F}=0$.

**Figure 9 nanomaterials-11-01219-f009:**
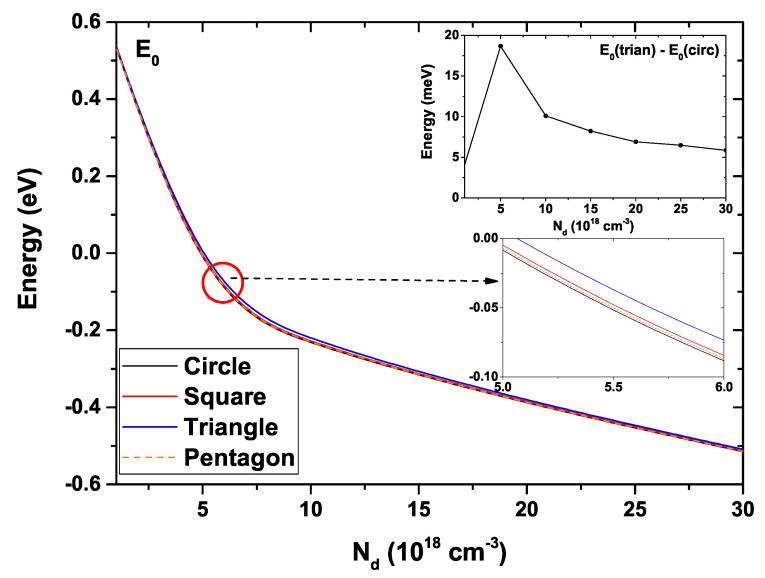
Ground state energy for each GaAs quantum wire structure as a function of the $N_{d}$ parameter. The inset shows the energy difference between the triangular and circular wires. Calculations are for $R_{0}=30$ nm and T=10 K.

**Figure 10 nanomaterials-11-01219-f010:**
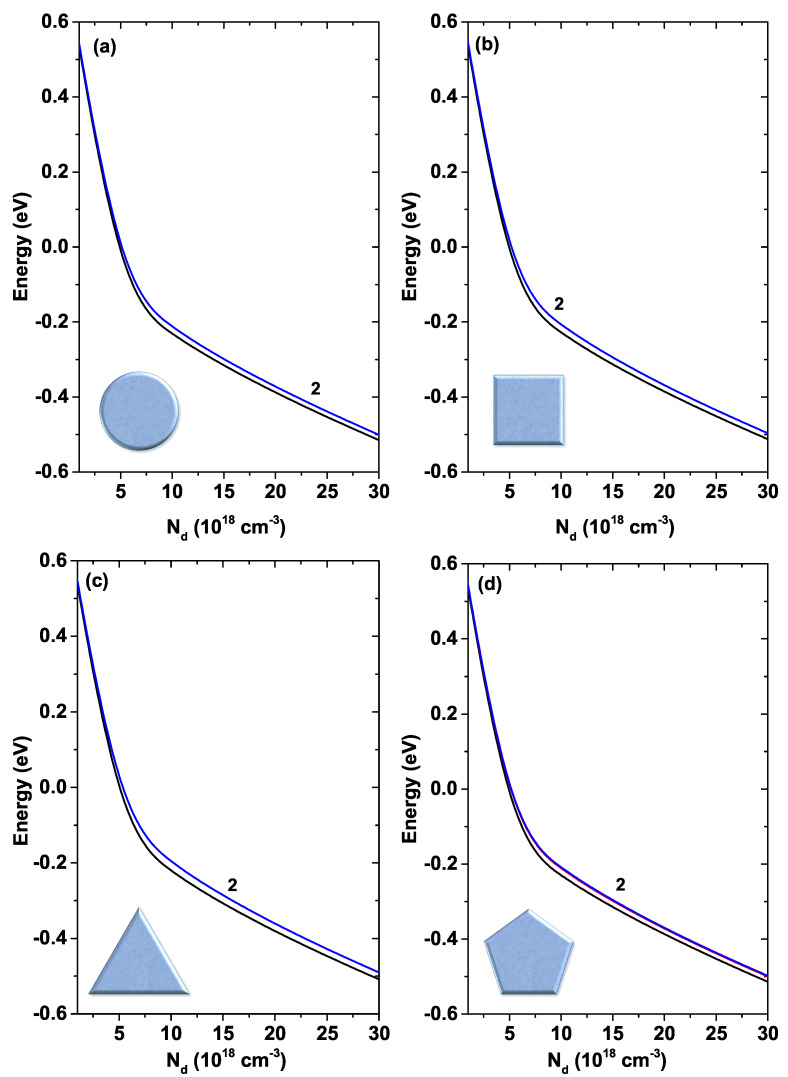
First two states of each GaAs quantum wire system as a function of the $N_{d}$ parameter. The label 2 in each figure indicates that the first excited state is doubly degenerated. Calculations are for $R_{0}=30$ nm and T=10 K.

**Figure 11 nanomaterials-11-01219-f011:**
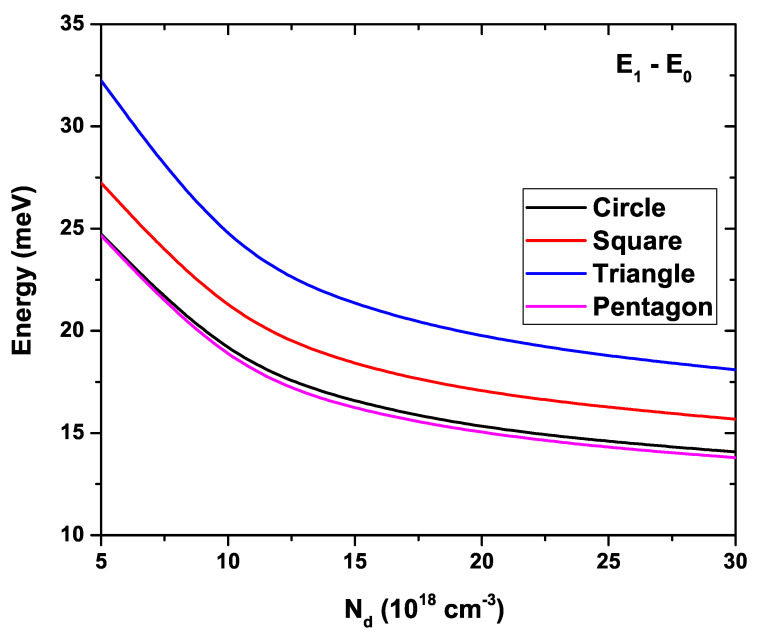
Energy difference between the first excited state and the ground state for each GaAs quantum wire system as a function of the $N_{d}$ parameter. Calculations are for $R_{0}=30$ nm and T=10 K.

**Figure 12 nanomaterials-11-01219-f012:**
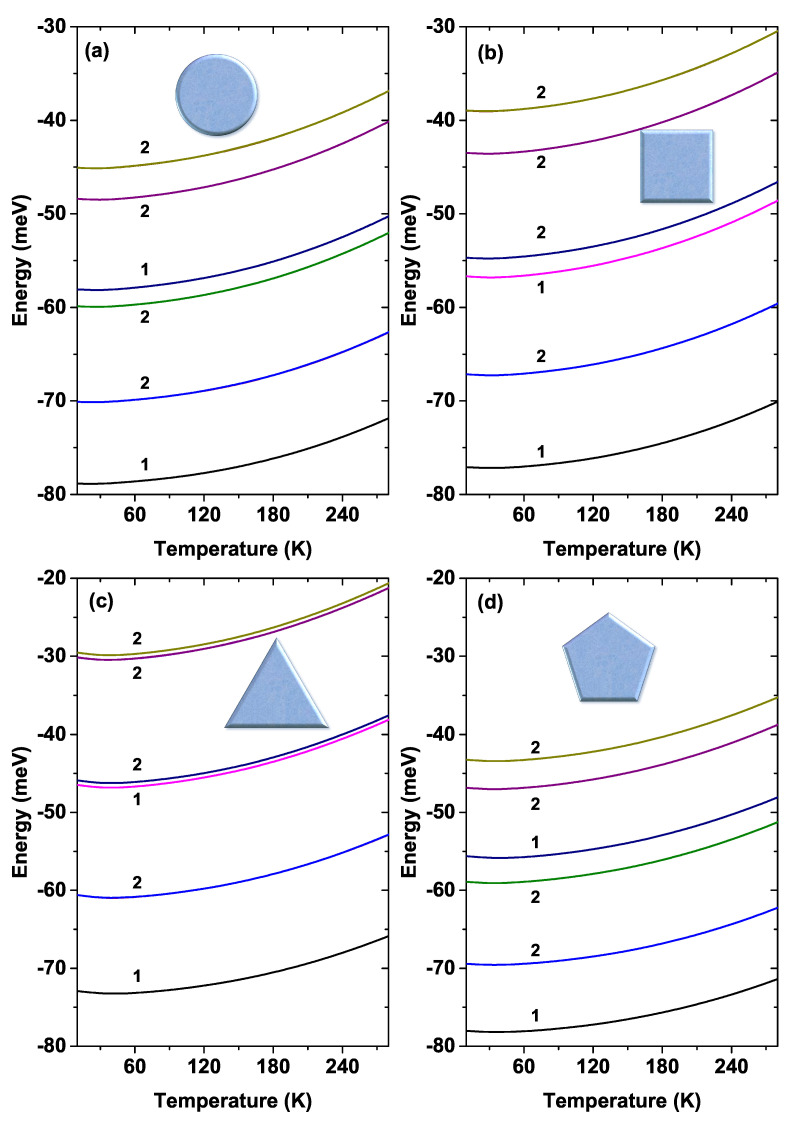
The first six energy levels of each GaAs quantum wire system as a function of temperature. The labels 1 and 2 indicate the degree of degeneracy. Calculations are for $R_{0}=50$ nm and $N_{d}=2\times 10^{18}$ cm${}^{-3}$.

**Figure 13 nanomaterials-11-01219-f013:**
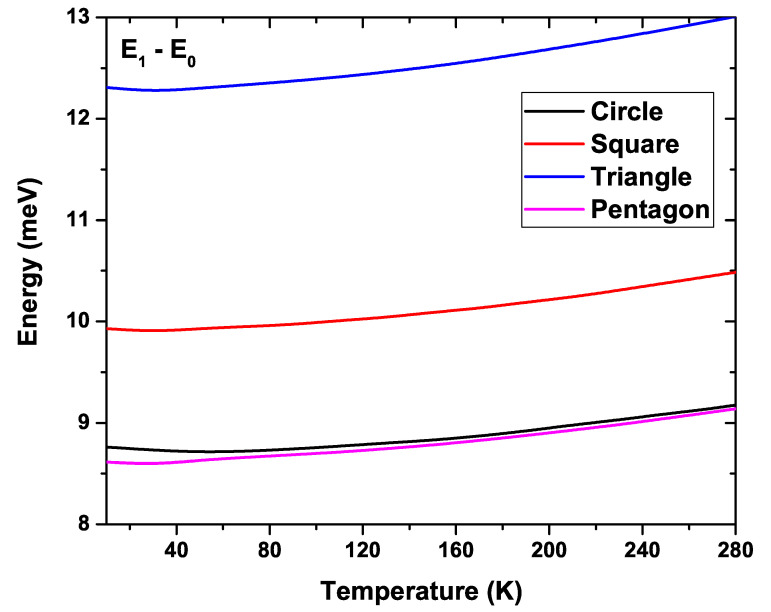
Energy difference between the first excited state energy and the ground state energy for each GaAs quantum wire system as a function of temperature. Calculations are for $R_{0}=50$ nm and $N_{d}=2\times 10^{18}$ cm${}^{-3}$.

**Figure 14 nanomaterials-11-01219-f014:**
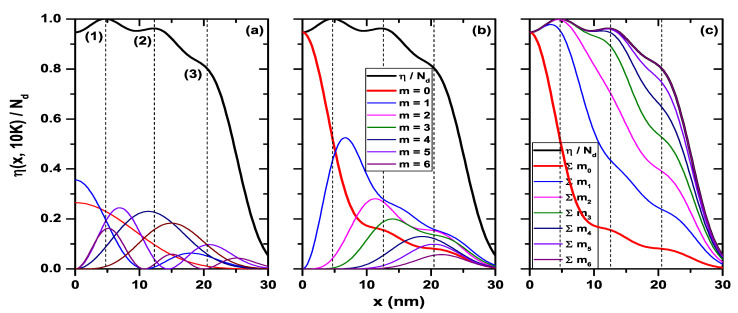
Electron density for circular GaAs quantum wire. In (**a**) the $|\psi_{i}{\left(x,y=0\right)|}^{2}$ states that contribute with the highest percentage to the density profile in points denoted by (1), (2), and (3) to the density profile. In (**b**), the contribution to the electron density for each quantum number *m*. In (**c**), the sum over the states with equal *m*. Note that in the case of the circular quantum wire, and due to its axial symmetry, the $\psi_{i}\left(x,y\right)$ wave function in Equation (Equation 6) can be written as $\psi_{i}\left(x,y\right)=f_{l}\left(\rho \right)\;\exp\left(im\phi \right)$, where m=0,±1,±2,... is the azimuthal quantum number, l=1,2,3,... is the radial quantum number, and $\rho =\sqrt{x^{2}+y^{2}}$. Calculations are for $R_{0}=50$ nm, $N_{d}=2\times 10^{18}$ cm${}^{-3}$, and T=10 K.

**Figure 15 nanomaterials-11-01219-f015:**
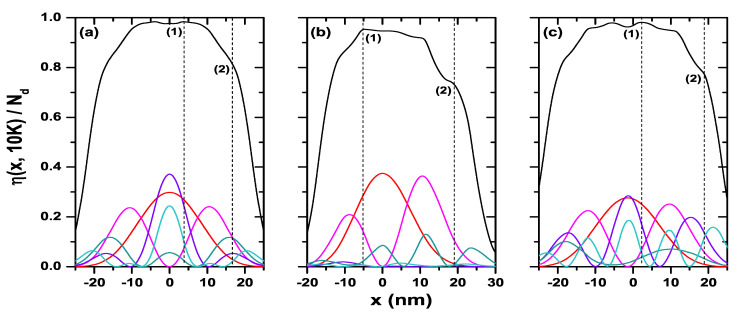
Electron density and $|\psi_{i}{\left(x,y=0\right)|}^{2}$ that contribute with the highest percentage to the density profile in points (1) and (2). Results are for square (**a**), triangle (**b**), and pentagon (**c**) QW. Calculations are for $R_{0}=50$ nm, $N_{d}=2\times 10^{18}$ cm${}^{-3}$, and T=10 K.

**Table 1 nanomaterials-11-01219-t001:** Contribution in percentage of each of the states to the oscillations at points (1), (2) and (3) in the density profile presented in Figure 14.

(m,l)	P${}_{1}$ (%)	P${}_{2}$ (%)	P${}_{3}$ (%)
(0,1)	23.0	10.2	1.5
(0,2)	19.6	0.7	6.5
(0,3)	8.6	5.5	
(1,1)	8.8	23.5	7.4
(1,2)	19.2	4.3	12.1
(1,3)	16.0	1.8	0.2
(2,1)	0.9	16.3	12.0
(2,2)	3.4	12.1	7.0
(3,1)	0.2	8.4	14.9
(3,2)	0.3	10.6	2.0
(4,1)		3.5	15.0
(4,2)		1.5	
(5,1)		1.3	12.3
(6,1)		0.3	6.9

**Table 2 nanomaterials-11-01219-t002:** The first 16 states and their percentage contribution to the electron density profile at points (1) and (2) of Figure 15.

	Square		Triangle		Pentagon	
$\psi_{n}$	**P** ${}_{1}$ **(%)**	**P${}_{2}$ (%)**	**P${}_{1}$ (%)**	**P${}_{2}$ (%)**	**P${}_{1}$ (%)**	**P${}_{2}$ (%)**
$\psi_{0}$	27.3	3.4	30.4	1.7	25.5	1.1
$\psi_{1}$		0.4	0.1			
$\psi_{2}$	7.6	13.8	14.4	14.4	7.2	7.2
$\psi_{3}$			14.2	19.5	7.4	3.3
$\psi_{4}$	0.7	19.7	10.5	18.0		
$\psi_{5}$	24.6	6.6	0.6	0.6	16.9	17.2
$\psi_{6}$			8.3	24.4		
$\psi_{7}$	2.8	10.8		2.3	2.1	7.0
$\psi_{8}$	13.5	16.3				
$\psi_{9}$	3.4		2.1	10.2	13.8	24.6
$\psi_{10}$	2.4	14.1	2.4	1.4	3.9	4.1
$\psi_{11}$			0.4			
$\psi_{12}$			9.2	1.6	5.6	6.0
$\psi_{13}$	1.9	0.8	6.7	4.8		
$\psi_{14}$	10.2	2.9	0.7	1.1	5.1	15.9
$\psi_{15}$	1.8	8.3				

## Data Availability

No new data were created or analyzed in this study. Data sharing is not applicable to this article.

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
