# Peer review of "Self-Consistent Schrödinger-Poisson Study of Electronic Properties of GaAs Quantum Well Wires with Various Cross-Sectional Shapes"

_nanomaterials, 2021, doi:10.3390/nano11051219_

Round 1

Reviewer 1 Report

In this manuscript, the authors utilize a self-consistent solution approach --- coupled Schrodinger and Poisson equations --- based on the finite-element discretization to study the effect of geometry, donor density, and temperature on the electronic properties of quantum wires. The manuscript is well written and the work seems to have been carefully done. The authors should address the following comments prior to publication

1) The authors commonly used the self-consistent approach terminology throughout their manuscript. Adding a few more words that clarify which approach this refers to will help the readers. For instance, the self-consistent approach is commonly used in density functional theory (DFT), which is not what the paper employs.

2) It is not clear whether the wire being considered is finite or infinite. Based on the statements in the conclusions regarding finite wires, one would infer that they are infinite wires. If so, what boundary conditions are specified on the orbital along the infinite direction? Currently, it is just mentioned that is zero boundary condition, which is suitable for finite wires.

3) There are some recent works (https://doi.org/10.1103/PhysRevB.103.035101, https://doi.org/10.1103/PhysRevB.100.125143) that focus on the development and application of real-space techniques for the highly efficient solution of the Kohn-Sham DFT equations in cylindrical coordinates that are particularly useful in the study of quasi one dimensional systems, similar to those studied in the paper. Relating the current work, i.e.., advantages and disadvantages, to these previous works would significantly help the reader.

Author Response

The Referee:

In this manuscript, the authors utilize a self-consistent solution approach ---coupled Schrodinger and Poisson equations--- based on the finite-element discretization to study the effect of geometry, donor density, and temperature on the electronic properties of quantum wires. The manuscript is well written and the work seems to have been carefully done. The authors should address the following comments prior to publication

Our Reply:

We want to thank the Referee for his/her report which helped us to improve the quality of the manuscript.

The Referee:

1) The authors commonly used the self-consistent approach terminology throughout their manuscript. Adding a few more words that clarify which approach this refers to will help the readers. For instance, the self-consistent approach is commonly used in density functional theory (DFT), which is not what the paper employs.

Our Reply:

We want to thank the Referee for his/her comment. In paragraphs 1 and 2 in right-hand column of page 3, of the revised version of the manuscript, we have included some comments related with the self-consistent procedure. The theoretical framework deeply explains the procedure.

The Referee:

2) It is not clear whether the wire being considered is finite or infinite. Based on the statements in the conclusions regarding finite wires, one would infer that they are infinite wires. If so, what boundary conditions are specified on the orbital along the infinite direction? Currently, it is just mentioned that is zero boundary condition, which is suitable for finite wires.

Our Reply:

We want to thank the Referee for his/her comment. In the revised version of the manuscript, we have included Figure 1, where we present the quantum wire system on which the study is developed. The studied system is a quantum wire of infinite length with an infinite potential barrier in the radial direction. The wave functions vanish at the boundary of the quantum wire, consistent with the Dirichlet boundary conditions. The corresponding comment has been included in the theoretical framework. In the theoretical framework, we have included the equations that account for the free particle motion along the system's axial axis, which is consistent with the infinite length model.

The Referee:

3) There are some recent works (https://doi.org/10.1103/PhysRevB.103.035101, https://doi.org/10.1103/PhysRevB.100.125143) that focus on the development and application of real-space techniques for the highly efficient solution of the Kohn-Sham DFT equations in cylindrical coordinates that are particularly useful in the study of quasi one dimensional systems, similar to those studied in the paper. Relating the current work, i.e.., advantages and disadvantages, to these previous works would significantly help the reader.

Our Reply:

We want to thank the Referee for his/her comment. In the last paragraph in the left-hand column of page 2 of the revised version of the manuscript, we have included the two references suggested by the Referee with the corresponding comments. In the last paragraph of the Introduction section, we have included our comments related to our work's advantages and disadvantages concerning the previously published papers. We have improved the state of the art by including in the paragraph in the left column of page 2 a set of references related to our research.

We hope that the revised version of the manuscript will be suitable for publication in the Nanomaterials journal.

Reviewer 2 Report

The authors  investigate the electronic properties of quantum wires of electron gas with various shapes by numerically solving the coupled Schrodinger equation and Poission equations self-consistently. The shapes of quantum wire s are defined by the confining potentials, and they assume electrons are totally confined within the volume of quantum wires. Overall, the problems they have addressed are not new and the computational methods they have implemented are also old. I did not see any important advancing in knowledge of quantum wires and significant improvement in theoretical modelings. Therefore, I do not enthusiastically recommend the publication of the manuscript.

Author Response

The Referee:

The authors investigate the electronic properties of quantum wires of electron gas with various shapes by numerically solving the coupled Schrodinger equation and Poission equations self-consistently. The shapes of quantum wires are defined by the confining potentials, and they assume electrons are totally confined within the volume of quantum wires. Overall, the problems they have addressed are not new and the computational methods they have implemented are also old. I did not see any important advancing in knowledge of quantum wires and significant improvement in theoretical modelings. Therefore, I do not enthusiastically recommend the publication of the manuscript.

Our Reply:

We want to thank the Referee for his/her report, which helped us improve the manuscript's quality. We consider that the research we report here is novel and that it makes significant contributions to the study of 1D low-dimensional systems, known as quantum well wires. We have used the finite element method combined with a self-consistent calculation to solve the Schrödinger, Poisson, and charge neutrality equations that have been implemented, only since 2018, in the licensed software COMSOL-Multiphysics. This tool makes it possible to extend the study problem to quantum wires with finite confinement potential and subjected to the effects of electric and magnetic fields that preserve the system's symmetry. They are even allowed to model systems where fields break symmetries. We have extended the problem of quantum wires with a circular cross-section to other types of geometries, such as wires with a triangular, square, and pentagonal cross-section. This fact has allowed us to infer the importance and/or relevance of the geometric shape concerning the size. We have observed and explained in-depth detail the origin and fundamental characteristics of the oscillations in the electron density profile for structures with symmetry other than circular, a situation that had not been previously reported in the literature. Our research can be easily extended to quantum dots with multiple geometries and confining potentials.

All of the above leads us to think that our work is really novel and that it can be appreciated by researchers in the field of low-dimensional systems. We hope that we have provided sufficient arguments to the Referee for our article to be seen as suitable for publication in the Nanomaterials journal.

Reviewer 3 Report

Report on manuscript 1174643 Gil-Corrales:

In the manuscript entitled "Self-consistent Schrodinger-Poisson study of electronic properties in quantum-well wires" by Gil-Corrales and his/her coworkers, the authors presented a detailed numerical investigation on the electronic properties including wave functions, eigenstate energies, electronic potentials, and electron densities for quantum-well wires with different geometrical boundaries: Circle, Square, Triangle, and Pedangon. By following the method proposed in Ref. [40], the authors solved the Schrodinger equation and Poisson equation together in an iterative manner so that after certain steps the system can be eventually solved consistently. In this way, the authors performed a series of numerical simulations to show wave functions, self-consistent confining potentials, normalized electron density functions, ground and excited state eigenenergies. 

The results do show some interesting features. For example, the Friedel-like oscillations appeared in electron density profile, due to the degree of of occupation of internal and surface states. By comparing the critical quantities for four different geometrical boundaries, the authors found few interesting points on the electronic properties mentioned above. As such, the reported results might be of interest for other people working in this field.

After carefully going through the manuscript, I found the work is written and organized in a logical way. The content can also be followed without much difficulty. In addition, most of the presented results are technically correct.

However, when asked whether I would recommend it for publication in Nanomaterials, I do have the following reservations for the authors to consider:

(1)  As commented above, the work is to numerically study the electronic properties of a quantum-well wire. As such, a global picture of the researched problem might be difficult to approach. But nevertheless, it would be highly appreciated if the authors could qualitatively provide a such global picture to the problem under investigation. For example, I noticed that most of the simulations were performed at temperature of 10K. What will happen if the temperature increases?

(2)  Since the key messages drawn in this work are based upon the simulations, the complete information on parameter settings would be helpful for understanding the results. I notice that some of parameters were unknown for most of the figures presented in the manuscript. Please provide these parameter values in the revision.

(3)  It would be helpful if a proper coordinate system with scales is shown in Figure 1. Also here, I am sorry that I didn't picture the so-called s, p, and d shape. Could you please illustrate these in Figure 1 to guide the reader?

(4)  The x-y coordinates were lack of scales in Figure 2. Please amend it.

(5)  In Figures 6(a) and (b) as well as Figures 9(a) and (b), the x-axis scale is missing.

(6)  English issues:  There are many grammar issues and typos in the manuscript. Please polish the English and correct those typos.

In short, the work is interesting and could be publishable in Nanomaterials. However, before its formal acceptance, as pointed out above, there are some issues appearing in the current version. I welcome the authors to revise the manuscript by addressing the questions and comments raised above before a formal acceptance can be reached.

Author Response

The Referee:

In the manuscript entitled "Self-consistent Schrodinger-Poisson study of electronic properties in quantum-well wires" by Gil-Corrales and his/her coworkers, the authors presented a detailed numerical investigation on the electronic properties including wave functions, eigenstate energies, electronic potentials, and electron densities for quantum-well wires with different geometrical boundaries: Circle, Square, Triangle, and Pentagon. By following the method proposed in Ref. [40], the authors solved the Schrodinger equation and Poisson equation together in an iterative manner so that after certain steps the system can be eventually solved consistently. In this way, the authors performed a series of numerical simulations to show wave functions, self-consistent confining potentials, normalized electron density functions, ground and excited state eigenenergies.

The results do show some interesting features. For example, the Friedel-like oscillations appeared in electron density profile, due to the degree of occupation of internal and surface states. By comparing the critical quantities for four different geometrical boundaries, the authors found few interesting points on the electronic properties mentioned above. As such, the reported results might be of interest for other people working in this field.

After carefully going through the manuscript, I found the work is written and organized in a logical way. The content can also be followed without much difficulty. In addition, most of the presented results are technically correct.

Our Reply:

We want to thank the Referee for his/her report which helped us to improve the quality of the manuscript.

The Referee:

However, when asked whether I would recommend it for publication in Nanomaterials, I do have the following reservations for the authors to consider:

(1) As commented above, the work is to numerically study the electronic properties of a quantum-well wire. As such, a global picture of the researched problem might be difficult to approach. But nevertheless, it would be highly appreciated if the authors could qualitatively provide a such global picture to the problem under investigation. For example, I noticed that most of the simulations were performed at temperature of 10K. What will happen if the temperature increases?

Our Reply:

In the last paragraph of the Introduction section, we have included our comments related to our work's advantages and disadvantages concerning the previously published papers. We have improved the state of the art by including in the paragraph in the left column of page 2 a set of references related to our research.

In the revised version of the manuscript, we have included Figure 1, where we present the quantum wire system on which the study is developed. The studied system is a quantum wire of infinite length with an infinite potential barrier in the radial direction. The wave functions vanish at the boundary of the quantum wire, consistent with the Dirichlet boundary conditions. The corresponding comment has been included in the theoretical framework. In the theoretical framework, we have included the equations that account for the free particle motion along the system's axial axis, which is consistent with the infinite length model.

One of the objectives of this research is to explain the origin of the oscillations in the electron density profile, which are only present at low temperatures. However, as the Referee can see, for completions in Figs. 12 and 13 of the revised version of the manuscript, we include the effects of temperature on an electron's states confined in the quantum wires of study.

The Referee:

(2) Since the key messages drawn in this work are based upon the simulations, the complete information on parameter settings would be helpful for understanding the results. I notice that some of parameters were unknown for most of the figures presented in the manuscript. Please provide these parameter values in the revision.

Our Reply:

In the first paragraph of the Results and Discussion section, we have included the parameters used. We have substantially improved the presentation of all captions of the figures including full details on the parameters used in each of them.

The Referee:

(3) It would be helpful if a proper coordinate system with scales is shown in Figure 1. Also here, I am sorry that I didn't picture the so-called s, p, and d shape. Could you please illustrate these in Figure 1 to guide the reader?

Our Reply:

In Fig. 2 of the revised version of the manuscript, we have implemented the Referee's suggestion. We have added the scales in one of the figures, which are the same for all the other figures. In the third paragraph of the Results and Discussion section, we have included the details about the atomic-like orbitals.

The Referee:

(4) The x-y coordinates were lack of scales in Figure 2. Please amend it.

Our Reply:

We have amended Fig. 2, which is Fig. 3 in the revised version of the manuscript, by following the Referee's suggestion.

The Referee:

(5) In Figures 6(a) and (b) as well as Figures 9(a) and (b), the x-axis scale is missing.

Our Reply:

We have amended Figs. 6 and 9, which are Fig. 7 and 10 in the revised version of the manuscript, by following the Referee's suggestion.

The Referee:

(6) English issues: There are many grammar issues and typos in the manuscript. Please polish the English and correct those typos.

Our Reply:

We have carefully checked the writing and grammar of the manuscript.

The Referee:

In short, the work is interesting and could be publishable in Nanomaterials. However, before its formal acceptance, as pointed out above, there are some issues appearing in the current version. I welcome the authors to revise the manuscript by addressing the questions and comments raised above before a formal acceptance can be reached.

Our Reply:

We hope that the revised version of the manuscript will be suitable for publication in the Nanomaterials journal.